# *Exophiala chapopotensis* sp. nov., an extremotolerant black yeast from an oil-polluted soil in Mexico; phylophenetic approach to species hypothesis in the *Herpotrichiellaceae* family

**Martín R. Ide-Pérez[1], Ayixon Sánchez-Reyes[2]\*, Jorge Luis Folch-Mallol[1], María del Rayo Sánchez-Carbente[1]\***

**1** Centro de Investigación en Biotecnología, Universidad Autónoma del Estado de Morelos, Cuernavaca, Morelos, México, **2** Investigador por México-Instituto de Biotecnología, Universidad Nacional Autónoma de México, Cuernavaca, Morelos, México

\* ayixon.sanchez@ibt.unam.mx (ASR); maria.sanchez@uaem.mx (MRSC)

**Data Availability Statement:** The Whole Genome Shotgun project for LBMH1013 strain has been

## Abstract

*Exophiala* is a black fungi of the family *Herpotrichiellaceae* that can be found in a wide range of environments like soil, water and the human body as potential opportunistic pathogen. Some species are known to be extremophiles, thriving in harsh conditions such as deserts, glaciers, and polluted habitats. The identification of novel *Exophiala* species across diverse environments underlines the remarkable biodiversity within the genus. However, its classification using traditional phenotypic and phylogenetic analyses has posed a challenges. Here we describe a novel taxon, *Exophiala chapopotensis* sp. nov., strain LBMH1013, isolated from oil-polluted soil in Mexico, delimited according to combined morphological, molecular, evolutionary and statistics criteria. This species possesses the characteristic dark mycelia growing on PDA and tends to be darker in the presence of hydrocarbons. Its growth is dual with both yeast-like and hyphal forms. LBMH1013 differs from closely related species such as *E. nidicola* due to its larger aseptate conidia and could be distinguished from *E. dermatitidis* and *E. heteromorpha* by its inability to thrive above 37°C or 10% of NaCl. A comprehensive genomic analyses using up-to-date overall genome relatedness indices, several multigene phylogenies and molecular evolutionary analyzes using Bayesian speciation models, further validate its species-specific transition from all current *Exophiala*/Capronia species. Additionally, we applied the phylophenetic conceptual framework to delineate the species-specific hypothesis in order to incorporate this proposal within an integrative taxonomic framework. We believe that this approach to delimit fungal species will also be useful to our peers.

deposited at DDBJ/ENA/GenBank under the accession JAMFLB000000000 (https://www.ncbi.nlm.nih.gov/nuccore/JAMFLB000000000). The version described in this paper is version JAMFLB010000000.1. The genome assembly was deposited in the NCBI database under the BioProject ID PRJNA821518 (https://www.ncbi.nlm.nih.gov/bioproject?LinkName=nuccore_bioproject&from_uid=2283984903).

**Funding:** This work was partially funded by the Programa Presupuestario F003, grant number CF 2019 265222 Consejo Nacional de Humanidades Ciencia y Tecnología (CONAHCYT), granted to ASR. We thank to IBT-UNAM and the CONAHCYT program "Investigadoras e Investigadores por México, for supporting the Project 237. Also, CONAHCYT, México granted a scholarship (number: 779850) to MRIP. There was no additional external funding received for this study. The funders had no role in study design, data collection and analysis, decision to publish, or preparation of the manuscript.

**Competing interests:** The authors have declared that no competing interests exist.

## Introduction

The genus *Exophiala* and its type species (*E. salmonis*) were first described by Carmichael [1] and have been linked to cerebral mycetomas in fish and caused fatal epidemic infections in several trout and salmon hatcheries [2]. Importantly, immunocompromised humans can be also infected by *E. phaeomuriformis* and *E. dermatitidis* causing cutaneous and tracto-respiratory affections, among other *Exophiala* isolates that are opportunistic pathogens. Particularly, *Exophiala bergeri*, *E. dermatitidis*, *E. jeanselmei*, *E. lecanii-corni*, *E. mesophila*, *E oligosperma*, *E. spinifera*, and *E. xenobiotica* have been isolated from subcutaneous lesions and extreme kitchen environments such as dishwashers [3–5]. Other *Exophiala* representatives have as well been isolated from environmental samples polluted with hydrocarbons or other xenobiotics (*e. g E. macquariensis*, *E. frigidotolerans*, *E. exophialae*, *E. sideris* and *E. moniliae*) [6–10]. Recently, different *Exophiala* species and other black yeasts are proposed as organisms with high potential in bioremediation [7, 11].

Belonging to Chaetothyrialean fungi, *Exophiala* representatives are known for their dualism, its capacity to grow on alkylbenzenes as carbon source as well as their virulence towards animals [12, 13]. The discovery of *Exophiala* species in different environments suggests that the genus is highly biodiverse because of their metabolic adaptations, such as melanin and carotenoids synthesis, wall thickening and meristematic growth, dimorphism, thermo- and osmotolerance, adhesion, hydrophobicity, among others [14, 15]. These adaptations might allow *Exophiala* species to colonize a myriad of habitats and to tolerate stressful conditions such as low or high temperatures, limited water availability, high UV radiation, oligotrophic conditions, and presence of antibiotics (such as azoles) and xenobiotics such as polycyclic hydrocarbons [11, 16].

Despite the medical and environmental relevance of *Exophiala* the classification through classical phenotypic and phylogenetic analysis has been difficult. According to the Mycocosm database, there are 79 described species with legitimate nomenclature belonging to *Exophiala* genus. Nevertheless, since the description of new filamentous fungi is rising, the description and classification of novel species must be carried out in a way that encompass morphological, molecular, evolutive and statistics criteria, to avoid nomenclatural dualities and taxonomic chaos. The evaluation and application of a pragmatic species concept would be of vital importance to circumscribe adequately all *Exophiala* representatives and differentiate them from related groups *Capronia* or *Cladophialophora*.

Ide-Pérez et al., [17] have isolated two fungal strains LBMH1012 and LBMH1013, from an oil polluted site in Tabasco, Mexico, which by morphological and molecular criteria were classifiedinto the genus *Rhodotorula* and *Exopohiala*, respectively. Importantly, the *Rhodotorula* strain could remove monoaromatic hydrocarbons such as xylane, toluene and benzene, while the *Exophiala* sp. LBMH1013, removed up to 80% of monoaromatic and polyaromatic hydrocarbon, such as benzo-a-pyrene and phenanthrene simultaneously after 21 days of culture, emerging as a good candidate in bioremediation. Furthermore, in that moment, the strain LBMH1013 could not be classified with any recognized species within the genus, suggesting the possibility that it might be a novel species.

In the present work we describe a novel species of *Exophiala* genus, by which we propose to name *Exophiala chapopotensis* referring to the nature of the sample in which the specimen was collected as well as considering historical and cultural aspects of the isolation site. Since precolombian times in Mexico, fossil oil was known as "Chapopote", a derived náhuatl word "chapopotli" originated from "tzapotl" referring to an indigenous black and sweet fruit named "zapote" and "popoca" which means smoky. Therefore "chapopotli" indicates a black, shiny and smoky substance, which according to the phenotypic characteristic of the strain is an

acquired name [18], so the epithet *chapopotensis* derives from "chapopotli". We applied a phylophenetic approach to the species concept to delineate the species-specific hypothesis, which was subsequently tested under different molecular species delimitation methods; namely, a Bayesian implementation of Poisson tree processes (bPTP) [19] and the generalized mixed Yule-coalescent (GMYC) [20]. In order to incorporate this proposal within an integrative taxonomic framework, we offer a set of genomic, phylogenetic, and morpho-phenotypic evidence that supports our hypothesis.

## Materials and methods

### Isolation, culture and genomic sequencing of the LBMH1013 strain

The isolation of the strain LBMH1013 was introduced in previous publication and here is briefly recapitulated [17]. The sample was collected from contaminated soil sites in Santa Isabel, Cunduacán, Tabasco, México (the sampling site is located in a shared land and therefore controlled access is not required). Polluted soil samples were spread onto potato dextrose agar (PDA) plates supplemented with diesel (3%) and 100 µg/mL of kanamycin and ampicillin. The cultures were incubated for 20 days at 28˚C. Isolates were purified to axenic cultures and the strains that grew were selected using mineral medium supplemented with diesel (3%) and 10 ppm each of benzo [a] pyrene and phenanthrene as carbon sources. Physiological tests were performed with the API 20 NE system (according to the instructions of the manufacturer: bioMérieux, Marcy l'Etoile, France). The miniaturized system covers 20 tests for assimilation of carbon sources, fermentation and enzyme production. Interpretation of the results was done after 48 hours by visual inspection. We determined the strain growing rate on PDA at different temperatures (28, 35, 37 and 40˚C) for 20 days and in a pH range of 5–12 for three days. All tests were performed in triplicate.

For Whole Genome Shotgun (WGS) sequencing, the genomic DNA was extracted from the sample using the Quick-DNA HMW MagBead kit (Zymo Research catalogue number D6060). The extracted DNA was subjected to end-prep and adapter ligation with the native barcoding kit (EXP-NBD104) from Oxford Nanopore Technologies (ONT) following the manufacturer's instructions. The sequencing was performed using a MinION sequencer (ONT). The flow cell (R9.4.1) was primed using running buffer and library loading beads. The prepared library was loaded onto the flow cell and sequencing was performed for 24 hours. Base-calling was performed using Guppy v3.2.2 (ONT) on a high-performance computing (HPC) cluster. Quality control was performed using Nanoplot v1.33.0 (ONT) to assess the read length, quality and yield. The genome assembly was performed using Canu v2.2 [21]. The assembly was polished with proovframe v0.9.7 to improve the accuracy and correct frameshift errors [22]. Completeness was assessed with BUSCO v5.1.2 (Benchmarking Universal Single-Copy Orthologs) with Ascomycota as lineage option under the *Neurospora crassa* model. Gene prediction was executed with Augustus [23] and genome annotation was performed using KofamScan software (*exec_annotation* script v1.3.0) [24].

### Estimation of overall genome relatedness indices (OGRI)

In order to estimate the OGRI of LBMH1013 strain, we first selected all available genomic assemblies of the *Herpotrichiellaceae* family on the site https://www.ncbi.nlm.nih.gov/assembly/ (92 sequences, accessed on: 2022/09/27); with the following search details: "*Herpotrichiellaceae*"[Organism] AND (latest[filter] AND all[filter] NOT anomalous[filter]). Subsequently, we evaluated the mutational genomic distance (**D**) using the Mash program *v2.3* [25]. The average nucleotide identity (ANI) was calculated using FastANI *v1.33* [26]. The average amino acid identity analyses (AAI) and the percentage of conserved proteins (POCP), were

analyzed using CompareM *v0.023* (https://github.com/dparks1134/CompareM) and POCP calculator [27] respectively (based on the amino acid sequences predicted by Augustus [23]). Finally, the *hexa* nucleotide frequency analysis was performed with the Focus software with an updated database for fungal family *Herpotrichiellaceae* [28].

## Phylogenetic analysis

We explored the phylogenetic hypothesis using three different approaches, Multi-Locus Sequence Typing (MLST) with the SSU (accession OR035765.1), ITS (accession MT268970.1), LSU (accession OQ996257.1) and TUB2 (S1 Table) gene sequences retrieved from the Gen-Bank database (https://www.ncbi.nlm.nih.gov/genbank/). For this analysis, 59 CBS strains of the genus *Exophiala* and three strains of *Capronia* genus were used, with *Cyphellophora oxyspora* CBS698.73 as the outgroup (Table 1). The sequences for partial SSU, LSU, TUB2 in the strain LBMH1013 were deduced from the genome reported in this paper. The sequences corresponding to each gene were aligned with MUSCLE *v3.8.1551* [29] and every alignment was cured by the trimAl *v1.4* program with the *gappyout* option [30]. Subsequently, all individual alignments were concatenated in the multiplatform SEAVIEW *v5.0.5* [31]. Multi locus phylogenetic tree was reconstructed using the maximum likelihood method in IQ-TREE software multicore version 1.6.12 with SH-like approximate likelihood ratio test (SH-aLRT) for assess branch support. Also, two phylogenomic approaches were conducted, the alignment-free procedure implemented in JolyTree *v.1.1b.191021ac* from assemblies or draft genomes [32]; and the alignment-aware method implemented in the Universal Fungal Core Genes (UFCG) database and pipeline for fungal genome-wide analysis, by predicting single-copy orthologs highly conserved and inferring a phylogenenomic species tree [33]. The 92 genomic assemblies of the *Herpotrichiellaceae* family stated previously were used as input for JolyTree under default options, for UFCG the inputs were the proteins sequences generated from Augustus procedure to generate predicted proteomes from genome assemblies.

## Species delimitation by Bayesian Poisson Tree Processes (bPTP) and Generalized Mixed Yule Coalescent (GMYC) models

We conducted species delimitation tests using the statistical framework implemented in bPTP *version 0.51* [19] available on https://github.com/zhangjiajie/PTP and GMYC [34] available on https://github.com/iTaxoTools/GMYC-pyqt5. We made the ultrametric trees -to run GMCY- using the chronos function in APE package [35]. The newick and ultrametric trees generated from previous phylogenetic analysis (Multilocus, JolyTree and UFCG) were used as inputs on independent runs. Markov Chain Monte Carlo (MCMC) were set to $10^6$ generations as empirical evidence confirms the equilibrium distribution at this number. MCMC sampling interval–thinning and burn-in proportion were set as default. Convergence was visually analysed by checking the Posterior Log likelihood trace plot for every run. The complete pipeline to test the phylophenetic species concept in fungal genomes is available on https://github.com/ayixon/Fast-Fungal-Genome-Classifier.

## Data availability

The Whole Genome Shotgun project for LBMH1013 strain has been deposited at DDBJ/ENA/GenBank under the accession JAMFLB000000000. The version described in this paper is version JAMFLB010000000.1. The genome assembly was deposited in the NCBI database under the BioProject ID PRJNA821518.

**Table 1. Species and GenBank accession numbers of sequences used for multiple gene phylogenetic analysis in this study.**[T] **represents ex-type cultures.**

| Species | Strain | GenBank accession numbers | | | |
|---------|--------|------|------|------|------|
| | | ITS | LSU | SSU | TUB2 |
| *Capronia coronata* | CBS 617.96[T] | NR154745 | – | – | JN1124221 |
| *Capronia fungicola* | CBS 614.96[T] | KY484990 | – | NG058761 | – |
| *Capronia mansonii* | CBS 101.67[T] | AF050247 | AY004338 | X79318 | – |
| *Exophiala eucalyptorum* | CBS 121638[T] | NR132882 | KC455258 | KC455302 | KC455228 |
| *Exophiala abietophila* | CBS:145038[T] | MK442581 | NG066323 | – | – |
| *Exophiala alcalophila* | CBS 520.82[T] | JF747041 | AF361051 | JN856010 | JN112423 |
| *Exophiala angulospora* | CBS 482.92[T] | JF747046 | KF155190 | JN856011 | JN112426 |
| *Exophiala aquamarina* | CBS 119918[T] | JF747054 | – | JN856012 | JN112434 |
| *Exophiala arunalokei* | NCCPF106033 | MW724320 | – | – | – |
| *Exophiala asiatica* | CBS 122847[T] | NR111332 | – | – | – |
| *Exophiala attenuata* | F10685 | KT013095 | KT013094 | – | – |
| *Exophiala bergeri* | CBS 353.52[T] | EF551462 | FJ358240 | FJ358308 | EF551497 |
| *Exophiala bonariae* | CCFEE 5792 | JX681046 | KR781083 | – | – |
| *Exophiala brunnea* | CBS 587.66[T] | JF747062 | KX712342 | JN856013 | JN112442 |
| *Exophiala campbellii* | NCPF2274 | LT594703 | LT594760 | – | – |
| *Exophiala cancerae* | CBS 120420[T] | JF747064 | – | – | JN112444 |
| *Exophiala capensis* | CBS 128771[T] | JF499841 | MH876538 | – | – |
| *Exophiala castellanii* | CBS 158.58[T] | JF747070 | KF928522 | JN856014 | KF928586 |
| *Exophiala cinerea* | CGMCC 3.18778[T] | MG012696 | MG197820 | MG012724 | MG012745 |
| *Exophiala clavispora* | CGMCC:3.17512[T] | KP347940 | MG197829 | MG012733 | KP347931 |
| *Exophiala crusticola* | CBS 119970[T] | AM048755 | KF155180 | KF155199 | – |
| *Exophiala dermatitidis* | CBS 207.35[T] | AF050269 | KJ930160 | – | KF928572 |
| *Exophiala ellipsoidea* | CGMCC:3.17348[T] | KP347955 | KP347956 | KP347965 | KP347921 |
| *Exophiala embothrii* | CBS:146560 | MW045819 | MW045823 | – | – |
| *Exophiala equina* | CBS 119.23[T] | JF747094 | – | JN856017 | JN112462 |
| *Exophiala eucalypti* | CPC:27630 | KY173411 | KY173502 | – | – |
| *Exophiala exophialae* | CBS 668.76[T] | AY156973 | KX822326 | KX822287 | EF551499 |
| *Exophiala frigidotolerans* | CBS 146539 | LR699566 | LR699567 | – | – |
| *Exophiala halophila* | CBS 121512[T] | JF747108 | – | JN856015 | JN112473 |
| *Exophiala heteromorpha* | CBS:232.33[T] | MH855419 | MH866871 | – | – |
| *Exophiala hongkongensis* | CBS131511[T] | JN625231 | – | – | JN625236 |
| *Exophiala italica* | MFLUCC160245 | KY496744 | KY496723 | KY501114 | – |
| *Exophiala jeanselmei* | CBS 507.90[T] | AY156963 | FJ358242 | FJ358310 | EF551501 |
| *Exophiala lacus* | FMR 3995 | KU705830 | KU705847 | – | – |
| *Exophiala lavatrina* | NCPF:7893[T] | LT594696 | LT594755 | – | – |
| *Exophiala lecanii-corni* | CBS 123.33[T] | AY857528 | FJ358243 | FJ358311 | – |
| *Exophiala lignicola* | CBS:144622 | MK442582 | MK442524 | – | – |
| *Exophiala macquariensis* | CBS 144232 | MF619956 | – | – | MH297438 |
| *Exophiala mali* | CBS:146791[T] | MW175341 | MW175381 | – | - |
| *Exophiala mesophila* | CBS 402.95[T] | JF747111 | KX712349 | JN856016 | JN112476 |
| *Exophiala moniliae* | CBS 520.76[T] | KF881967 | KJ930162 | – | – |
| *Exophiala nagquensis* | CGMCC:3.17284 | KP347947 | MG197838 | MG012742 | KP347922 |
| *Exophiala nidicola* | FMR 3889 | MG701055 | MG701056 | – | – |
| *Exophiala nigra* | CBS 535.94[T] | KY115191 | KX712353 | – | – |

*(Continued)*

**Table 1.** (Continued)

| Species | Strain | GenBank accession numbers | | | |
|---------|--------|------|-----|-----|------|
| | | ITS | LSU | SSU | TUB2 |
| *Exophiala nishimurae* | CBS 101538[T] | AY163560 | KX822327 | KX822288 | JX482552 |
| *Exophiala oligosperma* | CBS 725.88[T] | AY163551 | KF928486 | FJ358313 | EF551508 |
| *Exophiala opportunistica* | CBS 109811[T] | JF747123 | KF928501 | – | JN112486 |
| *Exophiala palmae* | CMRP1196[T] | KY680434 | KY570929 | – | KY689829 |
| *Exophiala phaeomuriformis* | CBS 131.88[T] | AJ244259 | – | – | – |
| *Exophiala pisciphila* | CBS 537.73 | NR121269 | AF361052 | JN856018 | JN112493 |
| *Exophiala placitae* | CBS:121716 | MH863143 | MH874694 | – | – |
| *Exophiala polymorpha* | CBS 138920[T] | KP070763 | KP070764 | – | – |
| *Exophiala prostantherae* | CPC 38251[T] | MW175344 | MW175384 | – | – |
| *Exophiala pseudooligosperma* | YMF 1.6741 | MW616557 | MW616559 | MW616558 | MZ127830 |
| *Exophiala psychrophila* | CBS 191.87[T] | JF747135 | – | JN856019 | JN112497 |
| *Exophiala quercina* | CPC:33408[T] | MT223797 | MT223892 | – | – |
| *Exophiala radicis* | P2772 | KT099203 | KT723447 | KT723452 | KT723462 |
| *Exophiala salmonis* | CBS 157.67[T] | AF050274 | AY213702 | JN856020 | JN112499 |
| *Exophiala sideris* | CBS:121818[T] | HQ452311 | – | HQ441174 | HQ535833 |
| *Exophiala spinifera* | CBS 899.68[T] | AY156976 | – | – | EF551516 |
| *Exophiala tremulae* | CBS129355[T] | FJ665274 | – | KT894147 | KT894148 |
| *Exophiala xenobiotica* | CBS:128104 | MH864829 | MH876272 | – | – |
| **Exophiala chapopotensis** | **EXF-16016** | **MT268970** | **OQ996257** | **OR035765** | **-\*** |
| *Cyphellophora oxyspora* | CBS 698.73[T] | KC455249 | KC455262 | KC455305 | KC455232 |

**ITS**: internal transcribed spacer regions; **LSU**: 28S rDNA gene; **SSU**: 18S rDNA; **TUB2**: β-tubulin. *E. chapopotensis* data are indicated in bold. *The nucleotide sequence corresponding to β-tubulin is provided in the supplementary material.

## Nomenclature

The electronic version of this article in Portable Document Format (PDF) in a work with an ISSN or ISBN will represent a published work according to the International Code of Nomenclature for algae, fungi, and plants, and hence the new names contained in the electronic publication of a PLOS article are effectively published under that Code from the electronic edition alone, so there is no longer any need to provide printed copies.

In addition, new names contained in this work have been submitted to Fungal Names from where they will be made available to the Global Names Index. The unique Fungal Names number can be resolved and the associated information viewed through any standard web browser by appending the Fungal Names number contained in this publication to the prefix https://nmdc.cn/fungalnames/namesearch/toallfungalinfo?recordNumber=. The online version of this work is archived and available from the following digital repositories: LOCKSS.

## Results and discussion

### Genome assembly and genomic coherence estimators

The *Exophiala* sp. LBMH1013 genome sequencing yielded a total of 77,879 long-reads, with an average length of 9,168.70 bp and read N50 of 15,660.00 bp. The resulting genome assembly has an overall size of 27.8 Mb, which is comprised of 11 contigs with a contig N50 of 3.5 Mb. In addition, we successfully identified the mitochondrial genome with a size of 26,874 bp (Table 2). The assembly size agrees with the size observed in other strains of this genus (20~38

**Table 2. Genome assembly statistics of the strain LBMH1013.**

| Assembly metrics | |
|---|---|
| Genome size | 27.8 Mb |
| Number of organelles | 1 |
| Number of contigs | 11 |
| Contig N50 | 3.5 Mb |
| Contig L50 | 4 |
| GC percent | 51.5 |
| Genome coverage | 25.0x |
| Assembly level | Contig |
| GenBank assembly accession | GCA_024611085.1 |
| Taxon | *Exophiala* sp. LBMH1013 |
| WGS project | JAMFLB01 |
| Assembly type | haploid |
| Non-nuclear (Mitochondrion MT) size | 26,874 bp |
| Mitochondrion assembly accession | CM045182.1 |
| BUSCO Predictions | |
| Number of genes | 8383 |
| Busco completeness | 96.40% |
| Complete BUSCOs | 1645 |
| Complete and single-copy BUSCOs | 1645 |
| Complete and duplicated BUSCOs | 0 |
| Fragmented BUSCOs | 13 |
| Missing BUSCOs | 48 |
| Total BUSCO groups searched | 1706 |

Mb) [16, 36]; additionally, the assembly's completeness and lack of duplicated BUSCO genes strongly suggest that this is an haploid genome version with no evidence of contamination. The small number of fragmented or missing genes within the assembly indicates that it is a valuable resource for gene prediction and functional annotation efforts, with minimal loss of integrity. As a result, it can be anticipated that the rate of pseudogenes is low.

In order to investigate the taxonomic boundaries of the strain LBMH1013, we tested the genomic coherence hypothesis by estimating several genome metrics, namely, Mash genomic distance, ANI, AAI, POCP and kmers coherence. Under the general assumption that organisms of the same species share signatures of genomic coherence as a result of cohesive evolutionary forces; alternatively, in the speciation the genetic variation is significant enough to generate a transition in genomic coherence signatures. While LBMH1013 strain shares higher level of genomic similarity with representatives of the *Capronia* and *Exophiala* groups (Table 3), it is far from canonical coherence values (intraspecies thresholds ≥95% ANI) already studied in prokaryotes and eukaryotes for the species delineation problem [26, 37–45]. *Capronia coronata* CBS 617.96 and *Capronia epimyces* CBS 606.96 were the closest elements, with ~78% ANI, ~80% AAI and POCP values. *E. dermatitidis* CBS 120473 and CBS 109144 and *E. spinifera* JCM 15939 also exhibit comparable values, with differences of <1%. The OGRI metrics of the LBMH1013 strain are even far enough away from the *grey zone* values (90–94% ANI), which strongly supports the hypothesis of speciation as a new genomic context with significant genetic variation. Multiple studies have consistently demonstrated that the region between 90–94% of ANI encompasses intra- and inter-species, which can be interpreted as an active genomic transition region [26, 37, 44]. However, intraspecies relationships

**Table 3. Overall genome relatedness indices of the strain LBMH1013 against 92 representatives of the family *Herpotrichiellaceae*.** ANI, AAI, and POCP are expressed as percentages.

| Strain | Assembly accession | ANI | D | AAI | POCP |
|---|---|---|---|---|---|
| *Capronia coronata* CBS 617.96 | GCA_000585585.1 | 78.38 | 0.19 | 80.55 | 82.94 |
| *Capronia epimyces* CBS 606.96 | GCA_000585565.1 | 78.16 | 0.21 | 79.36 | 81.27 |
| *Exophiala dermatitidis* CBS 120473 | GCA_010883455.1 | 77.61 | 0.20 | 78.21 | 80.93 |
| *Exophiala dermatitidis* CBS 109144 | GCA_010883275.1 | 77.59 | 0.21 | 78.06 | 80.59 |
| *Exophiala dermatitidis* CBS 115663 | GCA_010883545.1 | 77.55 | 0.21 | 78.07 | 80.80 |
| *Exophiala dermatitidis* CBS 132754 | GCA_010883525.1 | 77.55 | 0.22 | 78.07 | 80.75 |
| *Exophiala phaeomuriformis* CBS 132758 | GCA_010883475.1 | 77.49 | 0.21 | 77.96 | 80.30 |
| *Exophiala dermatitidis* PF4406 | GCA_023621275.1 | 77.48 | 0.22 | 78.08 | 80.83 |
| *Exophiala dermatitidis* M20-04A | GCA_023621285.1 | 77.44 | 0.20 | 78.11 | 80.76 |
| *Exophiala dermatitidis* NIH/UT8656 | GCA_000230625.1 | 77.43 | 0.21 | 78.05 | 80.77 |
| *Exophiala dermatitidis* PKS1 | GCA_003349795.1 | 77.43 | 0.21 | 77.88 | 80.43 |
| *Exophiala spinifera* JCM 15939 | GCA_001599535.1 | 77.35 | 0.20 | 72.72 | 73.17 |
| *Exophiala dermatitidis* CBS 578.76 | GCA_010883425.1 | 77.26 | 0.22 | 78.07 | 80.57 |
| *Rhinocladiella mackenziei* IHM 22877 | GCA_001723215.1 | 75.15 | 0.26 | 74.54 | 69.90 |
| *Rhinocladiella mackenziei* CBS 650.93 | GCA_000835555.1 | 75.12 | 0.26 | 74.51 | 69.80 |
| *Rhinocladiella mackenzieid* H24460 | GCA_001723235.1 | 75.07 | 0.26 | 74.46 | 69.77 |
| *Exophiala oligosperma* A04 | GCA_015295565.1 | 73.34 | 0.23 | 72.73 | 73.15 |
| *Exophiala xenobiotica* CBS 118157 | GCA_000835505.1 | 73.25 | 0.23 | 72.64 | 73.39 |
| *Exophiala xenobiotica* CBS102455 | GCA_000798695.1 | 73.02 | 0.22 | 72.41 | 73.42 |
| *Exophiala sideris* CBS121828 | GCA_000835395.1 | 72.80 | 1.00 | 72.19 | 73.13 |
| *Exophiala* sp. JF 03-4F | GCA_022695825.1 | 72.71 | 1.00 | 72.10 | 74.54 |
| *Exophiala* sp. JF 03-3F | GCA_022695815.1 | 72.70 | 1.00 | 72.09 | 74.44 |
| *Cladophialophora bantiana* CBS 173.52 | GCA_000835475.1 | 72.48 | 0.30 | 71.87 | 71.42 |
| *Fonsecaea multimorphosa* CBS 102226 | GCA_000836435.1 | 72.48 | 1.00 | 71.87 | 71.15 |
| *Exophiala alcalophila* JCM 1751 | GCA_001599775.1 | 72.47 | 1.00 | 71.86 | 73.10 |
| *Fonsecaea monophora* CBS 269.37 | GCA_001642475.1 | 72.47 | 1.00 | 71.86 | 71.16 |
| *Fonsecaea multimorphosa* CBS 980.96 | GCA_001646985.1 | 72.47 | 1.00 | 71.86 | 71.14 |
| *Cladophialophora psammophila* CBS 110553 | GCA_000585535.1 | 72.46 | 0.30 | 71.85 | 70.87 |
| *Exophiala oligosperma* CBS72588 | GCA_000835515.1 | 72.43 | 0.22 | 71.82 | 71.56 |
| *Rhinocladiella similis* Poitiers_1 | GCA_024082115.1 | 72.40 | 0.26 | 71.79 | 71.86 |
| *Exophiala oligosperma* FKI-L8-BK-P1 | GCA_022813245.1 | 72.39 | 0.23 | 71.78 | 62.54 |
| *Exophiala calicioides* JCM 6030 | GCA_001599795.1 | 72.39 | 0.26 | 71.78 | 70.21 |
| *Fonsecaea pedrosoi* CBS 271.37 | GCA_000835455.1 | 72.32 | 1.00 | 71.71 | 70.88 |
| *Cladophialophora immunda* CBS110551 | GCA_000835495.1 | 72.31 | 0.30 | 71.70 | 68.01 |
| *Fonsecaea pugnacius* CBS 139214 | GCA_011800825.1 | 72.28 | 0.30 | 71.67 | 70.39 |
| *Fonsecaea pedrosoi* ATCC 46428 | GCA_020310725.1 | 72.20 | 1.00 | 71.59 | 70.90 |
| *Exophiala* sp. HKRS030 | GCA_023897205.1 | 72.16 | 0.22 | 71.55 | 71.97 |
| *Exophiala spinifera* BMU 08022 | GCA_010882995.1 | 72.11 | 0.30 | 71.50 | 72.18 |
| *Exophiala spinifera* CBS 116557 | GCA_010883385.1 | 72.10 | 0.30 | 71.49 | 72.52 |
| *Fonsecaea nubica* CBS 269.64 | GCA_001646965.1 | 72.08 | 0.26 | 71.47 | 71.16 |
| *Cladophialophora immunda* CBS110551 | GCA_000785585.1 | 72.07 | 0.26 | 71.46 | 68.12 |
| *Exophiala spinifera* CBS89968 | GCA_000836115.1 | 72.04 | 0.30 | 71.43 | 72.25 |
| *Fonsecaea erecta* CBS 125763 | GCA_001651985.1 | 72.03 | 0.26 | 71.42 | 71.20 |
| *Exophiala spinifera* BMU 00051 | GCA_010882955.1 | 72.03 | 0.30 | 71.42 | 72.11 |
| *Exophiala spinifera* BMU 00047 | GCA_010882975.1 | 72.03 | 0.30 | 71.42 | 72.62 |
| *Exophiala spinifera* CBS 126013 | GCA_010883305.1 | 72.02 | 0.30 | 71.41 | 72.56 |

*(Continued)*

**Table 3.** (Continued)

| Strain | Assembly accession | ANI | D | AAI | POCP |
|---|---|---|---|---|---|
| *Exophiala spinifera* CBS 101539 | GCA_010883435.1 | 72.02 | 0.30 | 71.41 | 72.64 |
| *Exophiala spinifera* CBS 131564 | GCA_010883335.1 | 72.00 | 0.30 | 71.39 | 72.24 |
| *Exophiala spinifera* CBS 123469 | GCA_010883315.1 | 71.99 | 0.30 | 71.38 | 72.21 |
| *Phialophora macrospora* BMU 00149 | GCA_016109925.1 | 71.38 | 0.30 | 70.77 | 70.88 |
| *Phialophora macrospora* BMU 07676 | GCA_016109975.1 | 71.37 | 0.30 | 70.76 | 71.08 |
| *Phialophora macrospora* BMU 07066 | GCA_016109505.1 | 71.36 | 0.30 | 70.75 | 71.00 |
| *Cladophialophora carrionii* CBS 160.54 | GCA_000365165.2 | 71.36 | 1.00 | 70.75 | 72.55 |
| *Phialophora macrospora* BMU 00115 | GCA_016109565.1 | 71.35 | 0.30 | 70.74 | 70.93 |
| *Phialophora macrospora* BMU 00106 | GCA_016109955.1 | 71.31 | 0.30 | 70.7 | 70.29 |
| *Phialophora chinensis* BMU 07637 | GCA_016110035.1 | 71.31 | 1.00 | 70.7 | 69.39 |
| *Phialophora chinensis* BMU 07630 | GCA_016109575.1 | 71.30 | 1.00 | 70.69 | 69.59 |
| *Phialophora chinensis* BMU 07661 | GCA_016110055.1 | 71.30 | 1.00 | 70.69 | 69.51 |
| *Phialophora americana* BMU 07696 | GCA_016110225.1 | 71.29 | 0.30 | 70.68 | 69.11 |
| *Cladophialophora carrionii* KSF | GCA_001700775.1 | 71.28 | 1.00 | 70.67 | 72.70 |
| *Phialophora chinensis* BMU 07664 | GCA_016109625.1 | 71.27 | 1.00 | 70.66 | 69.62 |
| *Phialophora expanda* BMU 02323 | GCA_016109585.1 | 71.26 | 1.00 | 70.65 | 70.28 |
| *Phialophora chinensis* BMU 07609 | GCA_016110015.1 | 71.26 | 1.00 | 70.65 | 69.73 |
| *Capronia semiimmersa* CBS27337 | GCA_000835435.1 | 71.25 | 0.30 | 70.64 | 71.40 |
| *Phialophora americana* BMU 01244 | GCA_016110145.1 | 71.25 | 0.30 | 70.64 | 68.96 |
| *Phialophora verrucosa* BMU 04928 | GCA_016109935.1 | 71.25 | 1.00 | 70.64 | 70.30 |
| *Phialophora expanda* BMU 01245 | GCA_016110025.1 | 71.25 | 1.00 | 70.64 | 70.21 |
| *Phialophora verrucosa* BMU 05960 | GCA_016109465.1 | 71.21 | 1.00 | 70.60 | 67.29 |
| *Phialophora expanda* BMU 09470 | GCA_016110005.1 | 71.21 | 1.00 | 70.60 | 70.21 |
| *Phialophora americana* BMU 06000 | GCA_016110105.1 | 71.20 | 0.30 | 70.59 | 69.02 |
| *Phialophora americana* BMU 07645 | GCA_016110215.1 | 71.20 | 0.30 | 70.59 | 69.05 |
| *Phialophora verrucosa* BMU 07712 | GCA_016109475.1 | 71.18 | 1.00 | 70.57 | 70.39 |
| *Phialophora verrucosa* BMU 07678 | GCA_016109485.1 | 71.18 | 1.00 | 70.57 | 70.30 |
| *Phialophora americana* BMU 07652 | GCA_016110205.1 | 71.17 | 0.30 | 70.56 | 68.89 |
| *Phialophora verrucosa* BMU07605 | GCA_002099365.1 | 71.15 | 1.00 | 70.54 | 70.28 |
| *Phialophora tarda* CBS 111589 | GCA_016109495.1 | 71.15 | 1.00 | 70.54 | 69.61 |
| *Phialophora americana* BMU 09530 | GCA_016110115.1 | 71.12 | 0.30 | 70.51 | 68.86 |
| *Phialophora americana* BMU 00125 | GCA_016110135.1 | 71.12 | 0.30 | 70.51 | 69.15 |
| *Cladophialophora yegresii* CBS 114405 | GCA_000585515.1 | 71.09 | 1.00 | 70.48 | 73.02 |
| *Exophiala aquamarina* CBS 119918 | GCA_000709125.1 | 71.06 | 1.00 | 70.45 | 68.41 |
| *Rhinocladiella mackenziei* B02 | GCA_015295605.1 | 70.60 | 1.00 | 69.99 | 35.17 |
| *Exophiala lecanii-corni* CBS 102400 | GCA_003955835.1 | 70.42 | 1.00 | 69.81 | 65.30 |
| *Capronia fungicola* CBS 614.96* | - | 70.03 | 1.00 | 69.78 | 71.93 |
| *Cladophialophora bantiana* | GCA_900092765.1 | 70.02 | 0.26 | 71.97 | 71.53 |
| *Exophiala* sp. S2_009_000R2a | GCA_004026505.1 | 70.00 | 1.00 | 69.39 | 58.79 |
| *Exophiala mesophila* CBS40295 | GCA_000836275.1 | 69.98 | 0.23 | 69.37 | 71.62 |
| *Exophiala mesophila* CCFEE 6314 | GCA_004011775.1 | 69.83 | 1.00 | 69.22 | 71.27 |
| *Exophiala mesophila* CBS120910 | GCA_000785215.1 | 69.74 | 1.00 | 69.13 | 70.67 |
| *Herpotrichiellaceae* sp. UM238 | GCA_000315175.1 | 62.82 | 1.00 | 62.21 | 61.80 |
| *Phialophora attinorum* CBS 131958 | GCA_001299255.1 | 61.85 | 0.30 | 61.24 | 58.19 |
| *Exophiala* sp. BO6 | GCA_015295625.1 | 61.41 | 0.30 | 60.80 | 26.28 |

*Assembly is available on: https://mycocosm.jgi.doe.gov/Capfu1/Capfu1.info.html

below ~80% ANI would be exceptionally rare, therefore, we argue that the LBMH1013 strain is a new genomospecies within this group since it does not show genomic coherence signatures with any of its described neighbours. AAI and POCP are consistent with this observations and unlike what was reported in the *Hypoxylaceae* family [46], the *Herpotrichiellaceae* members do not cluster around 70% ANI, since various representatives range from 61–69%; also the phenomenon of gene gain or loss does not appear to exert the most substantial influence on the speciation of the group, as indicated by the close margins observed among ANI, AAI, and POCP values for each taxon.

We present this analysis as an approximation to the genomic coherence hypothesis previously alluded by different groups [47, 48], or as a phenetic and simplified generalization of the model proposed by Steiner and Gregorius [49]. Although it is not possible to separate all the genomic representatives of the *Herpotrichiellaceae* family just with the genomo-phenetic criteria, it is possible to delineate taxa that stray from what we would consider coherence (the novel taxon, such as LBMH1013) (S1 Fig). A limitation of this analysis is the absence of universal thresholds for distinguishing species using genomic coherence estimators in eukaryotes. However, an increasing body of evidence demonstrates their utility in identifying trends, and successfully addressing the species delineation problem in prokaryotes, fungi, and other eukaryotes as mentioned before. A final piece of evidence on the genomic coherence hypothesis in LBMH1013 strain, was provided by the frequency of kmers analysis (*hexamers*) (S2 Table). This yielded *Capronia* and *Exophiala* as the most closely related groups (sharing ~57.03% of all kmers with *Capronia* and ~32.30% with *Exophiala* members). This suggest that the frequency of variants is quite distant from what we would expect among representatives of the same species ($\geq$90%) and therefore, a significant difference in the compositional regularity of bases that is characteristic in conspecific contexts. We conclude that strain LBMH1013 does not show global genomic consistency signatures with any sequenced species of the family *Herpotrichiellaceae*, and therefore hypothesized that it may be a new species of the *Exophiala* genus, in accordance with previous phylogenetic evidence obtained with the ITS gene [17].

## The phylogenetic hypothesis and species delimitation by bPTP and GMYC models

The phylophenetic species concept has been successfully applied in the demarcation of prokaryotic species and is closely related to the polyphasic approach to delimit microbial species [40, 47]. We believe this concept can be useful in the definition of new fungal species whenever its premises are evaluated under the corpus of integrative taxonomy [50]. For this, the monophyly hypothesis must also be tested under rigorous species delimitation models, such as those implemented in the bPTP and GMYC programs [19, 34]. In both methods, the speciation or coalescence are modelled as a function of number of substitutions or divergence time between and within species respectively. We have evaluated a robust multi-gene phylogeny, as well as two phylogenomic reconstructions under different principles (alignment-free distance-based and orthogroup based trees). Subsequently, we have subjected both three phylogenies to speciation tests and in all cases, we have obtained that the LBMH1013 strain is a new species. Due to the MLST tree containing a larger variety of distinct haplotypes in comparison to the phylogenomic reconstructions, and based on the observation that the strain LBMH1013 is predominantly grouped within a clade associated with *Exophiala*, we have concluded that *Exophiala* is the more appropriate genus for classifying the new species, rather than the sexual morph *Capronia*.

In the multi-gene tree, LBMH1013 clusters into an independent branch, separate from its nearest neighbours *E. heteromorpha* and *E. nidicola*, and constitutes a sister clade of the one formed by the species *E. dermatitidis*, *E. phaeomuriformis* and *C. mansoni* (Fig 1). Branches in both clades are strongly supported suggesting a congruent topology. The Bayesian speciation test

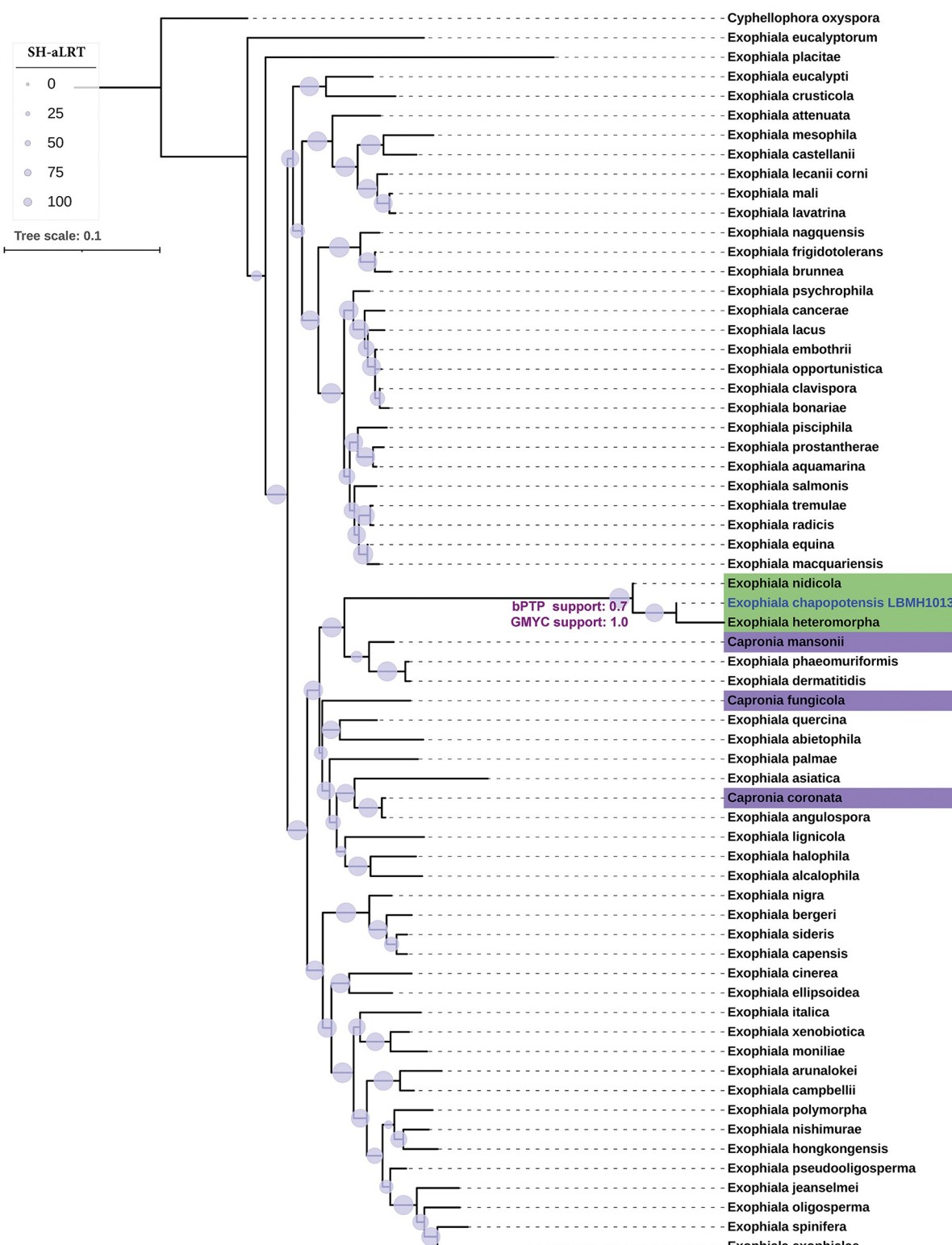

**Fig 1. Maximum-likelihood tree from concatenated sequences (SSU, ITS, LSU and B-TUB).** Branch supports are represented in nodes (periwinkle circles) as SH-like approximate likelihood ratio test (SH-aLRT) (%). Bar (0.1) represents number of changes per site. The tree was edited in iTOL online Version 6.7.6. bPTP and GMYC speciation partition supports for *E. chapopotensis* are depicted in violet on the corresponding branch. The indigo background corresponds to *Capronia* representatives. *Cyphellophora oxyspora* was used as the outgroup.

with bPTP and GMYC for this phylogenetic hypothesis delimited strain LBMH1013 in a separate partition (novel species) with Bayesian support of 0.7 and 1.0 respectively which favours the speciation hypothesis under both models and a flat prior. For bPTP test, the support values exhibit a robust correlation with the accuracy of the delimitation [19] as >60% of the partitions contain Bayesian supports >0.5 and in most species-specific prediction supports phylogeny.

Importantly, under these criteria the representatives of *Capronia* do not cluster in a monophyletic clade but rather show distinctive structure along the tree, intermingled with *Exophiala* species. This observation agrees well with other studies that support a cryptic phylogenetic delimitation between both genera using just rDNA sequences [51, 52]. The *Exophiala-Ramichloridium-Rhinocladiella* complex has been suggested to be polyphyletic [16, 52, 53], furthermore, *Capronia fungicola* did not exhibit consistency with its sister taxa, *C. epimyces* and *C. coronata* based on the previous OGRI analysis, raising the possibility of revising its classification in the light of these criteria.

The phylogenomic reconstructions with UFCG and JolyTree place strain LBMH1013 with *C. epimyces* and *C. coronata*, in a sister clade of *E. dermatitidis* (Figs 2 and 3). The topology of these trees is clearly different from that of the multigene tree because it contains a greater number of sites, and because not all taxa have a sequenced representative. For example, the *Exophiala* genus contains more available genomes, which may be due to its ecophenotypic attributes associated with clinical interest. More sequenced individual haplotypes are needed to assess whether *Capronia* is truly a monophyly-based clade. In these phylogenomic reconstructions, the Bayesian speciation tests concur to the partition where LBMH1013 is a new species with high statistical support, as suggested by the also highly supported phylogenetic topologies. The current classification of the genome GCA_015295605.1 *Rhinocladiella mackenziei* B02 (Fig 2, highlighted in yellow) is striking, which is not related to the *mackenziei* clade but corresponds to an independent lineage both phylogenetically and at the level of genomic measures. This assemblage deserves to be reclassified based on these observations. So, we considered that the pipeline used in the present study can be of interest for other authors.

## Micromorphology, phenotypic and transitive characteristic of LBMH1013

The strain LBMH1013 could grow as filamentous and yeast-like phenotypes, depending on the salt concentration (Fig 4A-4C). Notably, the strain demonstrated the ability to thrive in environments containing up to 6% diesel, as well as withstand concentrations of 10 ppm benzopyrene and phenanthrene [17]. The mycelium exhibited strong dark coloration in PDA and minimal medium (MM) supplemented with hydrocarbons. The micromorphology shows a septate mycelium, annellidic hyphae and production of conidia in PDA medium, also characteristics of asexual cycle and blastic conidiogenesis are deduced from microscopic observations (Fig 4D-4F). The strain LBMH1013 differs from the closely related *E. nidicola* by its larger aseptate conidia. The absence of growth at 40°C is a distinguishing character for LBMH1013, also at 37° the growth is severely disrupted [17], which is a distinctive feature of *E. dermatitidis* and *E. heteromorpha*. Additionally, LBMH1013 only tolerates ~5.84% NaCl, unlike *E. heteromorpha* in which viability has been observed at 10% [54]. Traditional taxonomic approaches to classify *Exophiala* have relied on morphological and phenotypic characteristics. However, in a scenario of increasing biodiversity, these methods have shown to lack sufficient resolution in the diagnosis of pleomorphic species with high microstructural similarities [55].

The phenotypic response on several carbon assimilation tests and growth response is shown in the Table 4. The strain LBMH1013 can grow efficiently on glucosamine, lactose and erythritol as sole carbon sources, while it shows limited growth on Yeast Nitrogen Base medium supplemented with NaNO$_3$ after 20 days (S2 Fig). The ability to grow on lactose is

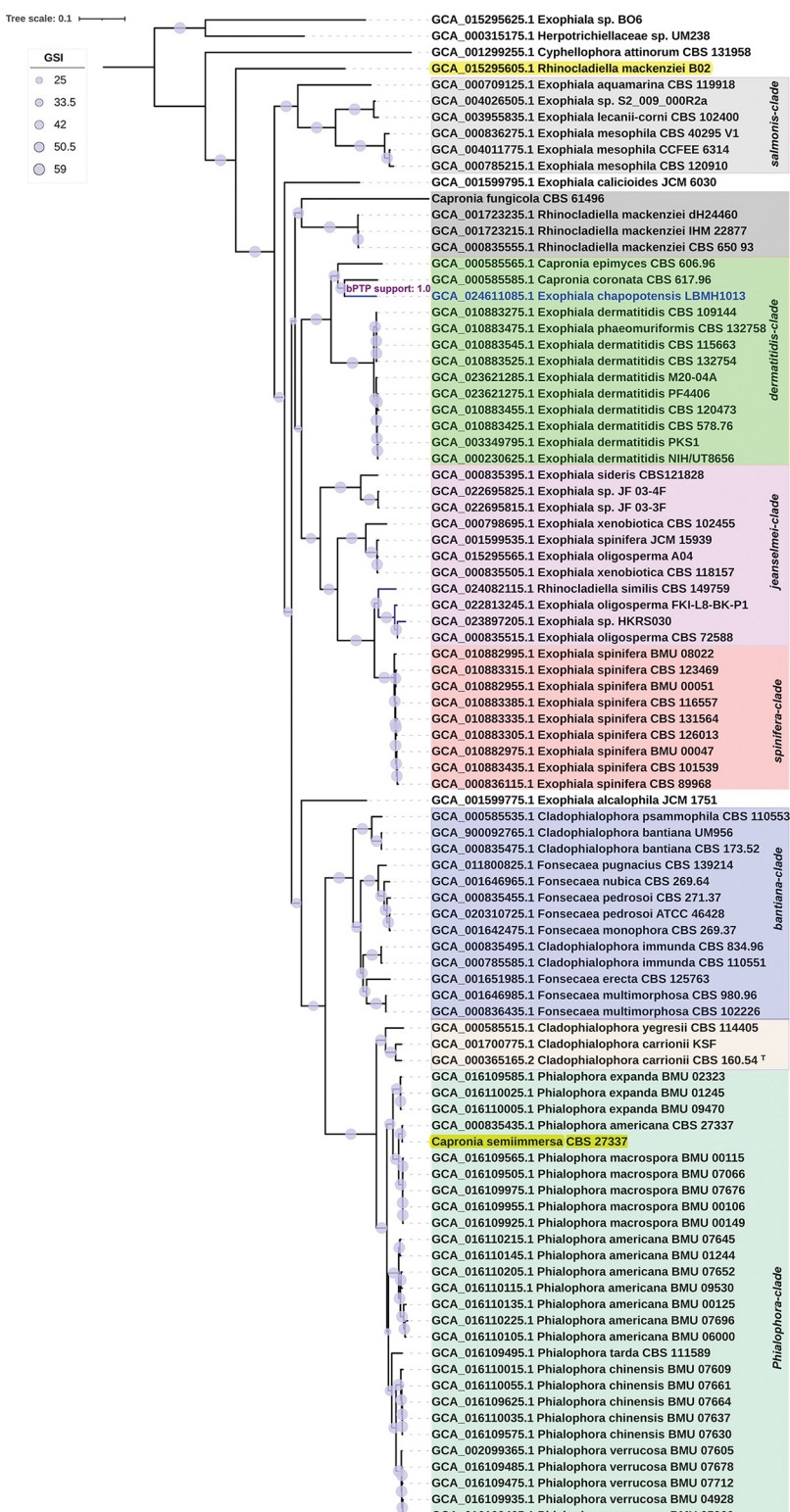

**Fig 2. UFCG tree from 59 concatenated fungal marker genes extracted from genomic sequences of the *Herpotrichiellaceae* assemblies.** The tree was rooted using the midpoint rooting method. Branches supports are represented by their Gene Support Index (GSI) values. The canonical monophyletic clades of the family identified by [16] are highlighted with coloured boxes. Bar (0.1) represents number of changes per site. The tree was edited in iTOL online version 6.7.6. bPTP speciation partition support for *E. chapopotensis* are depicted in violet on the corresponding branch. Contexts whose current nomenclature is incorrect are highlighted in yellow.

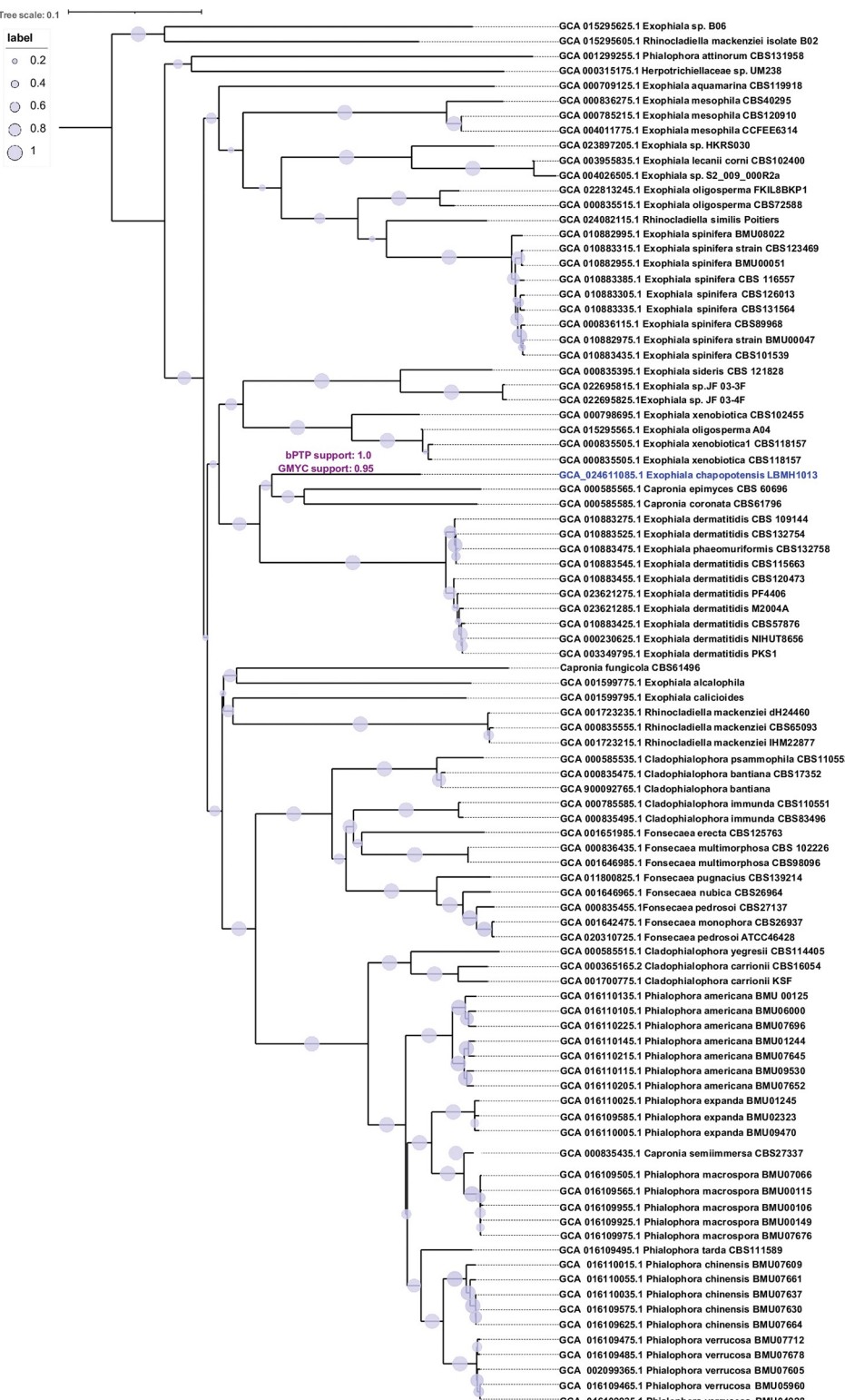

**Fig 3. Genomic distance-based phylogenetic trees from genome contig sequences of the *Herpotrichiellaceae* family.** The tree was rooted using the midpoint rooting method. The genome sequences accession is specified before each taxon name. Branch supports (0–1 scale) were assessed by JolyTree software. Bar (0.1) represents number of changes per site. The tree was edited in iTOL online version 6.7.6. bPTP and GMYC speciation partition supports for *E. chapopotensis* are depicted in violet on the corresponding branch.

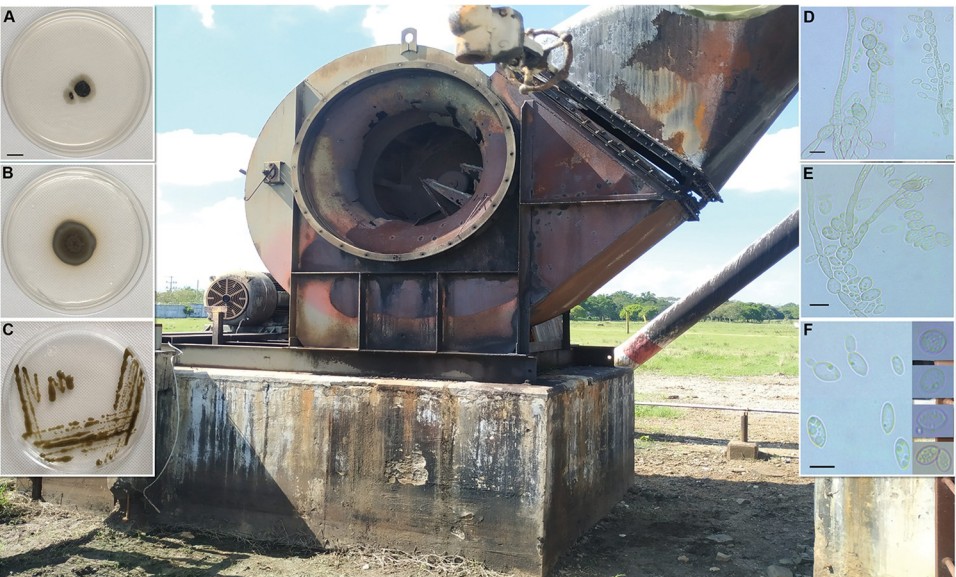

**Fig 4.** *Exophiala chapopotensis* **sp. nov (LBMH1013, Holotype). A** colony in MM with benzo [a] pyrene (100 ppm) after 1 week. **B, C** colonies on PDA after 1 week. **D** conidiogenous cell and septate hyphae, **E** multinucleated conidia, **F** aseptate conidia and budding cells. Scale bar: **A**-**C** 1 cm; **D, E** 10 μm. The background image corresponds to the Holotype isolation site, the installation of the flare stack for local oil duct, Cuanduacán, Tabasco, México is observed. The specimen was isolated from soil.

also a diagnostic criterion between *E. chapopotensis* and *E. heteromorpha*. The strain does not hydrolyze gelatin or ferment glucose, however it does produce urease, a widespread trait within the *Exophiala* genus, and hydrolyzes esculin. The assimilation was positive for (D-glucose, L-arabinose, D-mannose, D-mannitol, N-acetyl-glucosamine, D-maltose, Potassium gluconate; adipic acid and malic acid), while no assimilation was detected on capric acid, trisodium citrate and phenylacetic acid.

As opposed to *Capronia* which produces ascospores in sexual structures, in the strain LBMH1013 we have not observed sexual spores (just asexual conidia), which it is a distinguishing characteristic. Additionally, we have not observed evidence of a sexual cycle in the strain reported in this study. Furthermore, the strain LBMH1013 seems to be heterothallic, since the genome contains markers for only one of the MAT sexual idiomorphs (MAT1-1-4 and MAT1-1-1 (alpha-box)). The MAT1-1-4 homologous in LBMH1013 is located in the contig JAMFLB010000006.1), coordinates 120,461 bp-120,829 bp. The MAT1-1-1 homologous is located in the coordinates 121, 737 bp -122, 232 bp of the same contig as expected. As has been consistently shown in other studies, *Exophiala* representatives usually contain just one of the two *MAT* alleles, while *Capronia* representatives are homothalic [16], which constitutes a solid criterion for genetic differentiation between *Capronia* and *Exophiala*. In *E. dermatitidis* two idiomorphs (MAT1-1 and MAT1-2) have been detected, however, despite they are expressed, sexual cycle has not been found in the species [69].

In summary, this study presents a comprehensive description of the novel species *Exophiala chapopotensis* sp. nov., which was isolated from oil-polluted soil in Mexico. By employing a phylophenetic approach to the species concept, we rigorously tested hypotheses pertaining to genomic coherence, monophyly, and speciation using Bayesian Poisson Tree Processes (bPTP) and Generalized Mixed Yule Coalescent (GMYC) models. Also, key phenotypic differences were described that serve as diagnostic characters with other phylogenetically related strains. Collectively, these analyses provide robust support for the speciation hypothesis within

**Table 4. Growth, carbon utilization and micromorphology of *E. chapopotensis* LBMH1013 under several conditions\*.**

| | *E. chapopotensis* | *E. heteromorpha* | *E. dermatitidis* | *E. jeanselmei* | *E. viscosa* | *E. mesophila* |
|---|---|---|---|---|---|---|
| **Characteristics** | | | | | | |
| **Growth Temperature (˚C)** | | | | | | |
| 28 | + (optimum) | + | + | +(30 optimum) | + (23 optimum) | + (optimum) |
| 35 | d | + | + | + | ND | - |
| 37 | d/- | + | + | d/- | - | - |
| 40 | - | + | + | - | - | - |
| **Growth pH** | 5–12 | 2.5–10 | 5.4–8.1 | ND | ND | Up to 9.5 |
| **Growth in NaCl** | Up to 5.84% | Up to 10% | Up to 5% | Up to 9% | ND | Up to 10% |
| **Nitrate** | d | - | - | + | + | + |
| **Glucosamine** | + | ND | - | + | - | + |
| **Lactose** | + | - | d/- | v | - | - |
| **Erythritol** | + | ND | + | + | v | ND |
| **Glucose fermentation (GLU)** | - | - | + | - | ND | ND |
| **Urease (URE)** | + | - | + | + | + | ND |
| **Esculin hydrolysis (ESC)** | + | v | v | ND | + | ND |
| **Gelatin hydrolysis (GEL)** | - | ND | v | - | ND | ND |
| **D-glucose Assimilation** | + | + | + | + | + | + |
| **L-arabinose assimilation** | + | + | + | + | + | + |
| **D-mannose assimilation** | + | ND | + | + | ND | ND |
| **D-mannitol assimilation** | + | ND | + | + | v | ND |
| **N-acetyl-glucosamine** | + | + | v | + | + | + |
| **D-maltose assimilation** | + | v | v | + | + | ND |
| **Potassium gluconate assimilation** | + | ND | +/d | + | v | + |
| **Conidia And Conidiogenous Cell Micromorphology** | | | | | | |

| Species | Conidiogenous Cells | Size of Conidiogenous Cells (µm) | Conidia | Size of Conidia (µm) |
|---|---|---|---|---|
| *E. chapopotensis* | Terminal or lateral, cylindrical, elongated, ampulliform to lectiform | 8.2–11.9 x 3.2–5.5 | Oval, oblong to ellipsoid-shaped | 5.0–9 x 2.5–3.75 |
| *E. nidicola* | Intercalary, terminal or lateral, cylindrical, ellipsoidal or lageniform, annellidic, inconspicuous annellations | 9–12 × 2–3 (Intercalary cells) 5–9 × 2.5–4 (terminal and lateral) | Obovoidal to allantoid, hyaline, smooth, thin-walled | 3–5 x 1–1.5 |
| *E. heteromorpha* | Terminal or intercalary, occasionally lateral (free cells flask-shaped to elongate, rare annellations | 4.2–7.2 x 2.8–5.2 | Hyaline, thin-walled, broadly ellipsoidal | 2.6–4.2 x 1.6–2.5 |
| *E. dermatitidis* | Intercalary, cylindrical in main branches, broadly ellipsoidal to subglobose | 4–5 x 3.5 | Smooth, hyaline, thin-walled, broadly ellipsoidal | 2.6–4 x 2–3 |

\* The data corresponding to the strains compared in this table were extracted from the available literature [10, 54, 56–69]. + = good growth;— = no growth; d = poor growth; v = variable; ND: ambiguous or unknown; PNPG = β-galactosidase (Para-NitroPhenyl-ßDGalactopyranosidase).

this taxon phylogenetically related with human pathogens such as *E. heteromorpha* and *E. dermatitidis*. The environmental origin of *Exophiala chapopotensis* and its demonstrated capability to degrade aromatic hydrocarbons [17], underlines potential application in bioremediation efforts.

## Description

**Exophiala chapopotensis.   Ide-Pérez et al. 2023, sp. nov.** (Fig 4A-4F). Fungal Names no. FN 571584.

urn:lsid:nmdc.cn:fungalnames:571584

**Etymology:** *Exophiala chapopotensis* (cha.po.pot.en'sis. N.L. fem. adj. chapopotensis, referring to "Chapopote", a derived náhuatl word for heavy crude oil, material from where the type strain was first isolated).

**Holotype:** LBMH1013, isolated from petroleum contaminated soil in Santa Isabel, Cunduacán, Tabasco, México (18˚02'37.2" N, -93˚40'18.1" E, 10m altitude), October 2020, MR Ide-Pérez, preserved in glycerol 15% in Centro de Investigación en Biotecnología, Universidad Autónoma del Estado de Morelos.

**Isotype:** The strain is deposited in the Infrastructural Mycosmo Centre and Microbial Culture Collection Ex, Department of Biology, Faculty of Biotechnology, University of Ljubljana, Ljubljana, Slovenia. Deposit number: EXF-16016.

It can be distinguished from the closely related *E. nidicola* by its larger aseptate conidia; and from *E. dermatitidis* and *E. heteromorpha* because does not grow above 37˚C. Additionally, LBMH1013 only tolerates ~5.84% NaCl, while *E. heteromorpha* is viable up to 10%. Genomic and phylogenetic transitions support distinction from all other Exophiala/Capronia species. The growth is severed disrupted at 37˚C and not observed at or above 37˚C. Ovoid, oblong to ellipsoid-shaped and hyaline conidia, 5.0–9 μm x 2.5–3.75 μm, [n = 15]. The teleomorph is unknown.

## Supporting information

**S1 Table. Partial tubulin beta chain mRNA deduced from the genome of *Exophiala chapopotensis* LBMH1013.**
(PDF)

**S2 Table. Kmers frequency match between *Exophiala chapopotensis* LBMH1013 and its closest phylogenetic neighbours.**
(PDF)

**S1 Fig. Principal Components Analysis (PCA) performed on the representatives of the *Herpotrichiellaceae* family with genome metrics as factors.** Mash genomic distance, ANI, AAI, POCP. Analysis based on Correlations. Variances were computed as SS/N-1. Missing Data deletion: Casewise. No. of active Factors: 4; No. of active cases: 93. Eigenvalues: 2.84348 .830461 .314132. 011930. **NSP**: Non-Separable Data.
(PDF)

**S2 Fig. Growth of strain LBMH1013 in different carbon sources (colonies at 10 and 20 days of growth are shown) and at different pH (48 and 36 hours).**
(PDF)

## Acknowledgments

We thank to Dr. Alfonso Leija Salas and Dr. Salvador Barrera Ortiz for his help with the microscopy analysis.

## Author Contributions

**Conceptualization:** Ayixon Sánchez-Reyes, María del Rayo Sánchez-Carbente.

**Formal analysis:** Martín R. Ide-Pérez, Ayixon Sánchez-Reyes.

**Funding acquisition:** Ayixon Sánchez-Reyes, Jorge Luis Folch-Mallol, María del Rayo Sánchez-Carbente.

**Investigation:** Martín R. Ide-Pérez.

**Supervision:** Ayixon Sánchez-Reyes, María del Rayo Sánchez-Carbente.

**Writing – original draft:** Ayixon Sánchez-Reyes.

**Writing – review & editing:** Martín R. Ide-Pérez, Ayixon Sánchez-Reyes, Jorge Luis Folch-Mallol, María del Rayo Sánchez-Carbente.

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
