## [Decision Letter · Decision Letter 0]

5 Oct 2023

PONE-D-23-27467Exophiala chapopotensis sp. nov., an extremotolerant black yeast from an oil-polluted soil in Mexico; phylophenetic approach to species hypothesis in the Herpotrichiellaceae familyPLOS ONE

Dear Dr. Sánchez Reyes,

Thank you for submitting your manuscript to PLOS ONE. After careful consideration, we feel that it has merit but does not fully meet PLOS ONE’s publication criteria as it currently stands. Therefore, we invite you to submit a revised version of the manuscript that addresses the points raised during the review process.

We look forward to receiving your revised manuscript.

Kind regards,

Rajeev Singh

Academic Editor

PLOS ONE

Journal Requirements:

When submitting your revision, we need you to address these additional requirements. 1. Please ensure that your manuscript meets PLOS ONE's style requirements, including those for file naming. The PLOS ONE style templates can be found at https://journals.plos.org/plosone/s/file?id=wjVg/PLOSOne_formatting_sample_main_body.pdf and https://journals.plos.org/plosone/s/file?id=ba62/PLOSOne_formatting_sample_title_authors_affiliations.pdf 2. In your Methods section, please provide additional information regarding the permits you obtained for the work. Please ensure you have included the full name of the authority that approved the field site access and, if no permits were required, a brief statement explaining why. 3. We note that the grant information you provided in the ‘Funding Information’ and ‘Financial Disclosure’ sections do not match.  When you resubmit, please ensure that you provide the correct grant numbers for the awards you received for your study in the ‘Funding Information’ section. 4. Thank you for stating in your Funding Statement:  "This research was partially funded by the Programa Presupuestario F003, project code CF 2019 265222 (CONAHCYT, México), granted to ASR. The funders had no role in study design, data collection and analysis, decision to publish, or preparation of the manuscript." Please provide an amended statement that declares *all* the funding or sources of support (whether external or internal to your organization) received during this study, as detailed online in our guide for authors at http://journals.plos.org/plosone/s/submit-now.  Please also include the statement “There was no additional external funding received for this study.” in your updated Funding Statement. Please include your amended Funding Statement within your cover letter. We will change the online submission form on your behalf. 5. Thank you for stating the following in the Acknowledgments Section of your manuscript: "MR. Ide-Pérez was supported by Consejo Nacional de Humanidades Ciencia y Tecnología (CONAHCYT), México (Grant number: 779850). We thank to IBT-UNAM and the program “Investigadoras e Investigadores por México” from the Consejo Nacional de Humanidades Ciencia y Tecnología (CONAHCYT), México, for supporting the Project 237. This research was partially funded by the Programa Presupuestario F003, project code CF 2019 265222, granted to ASR. Also, we thank to Dr. Alfonso Leija Salas for his help with the microscopy analysis." We note that you have provided funding information that is not currently declared in your Funding Statement. However, funding information should not appear in the Acknowledgments section or other areas of your manuscript. We will only publish funding information present in the Funding Statement section of the online submission form. Please remove any funding-related text from the manuscript and let us know how you would like to update your Funding Statement. Currently, your Funding Statement reads as follows: "This research was partially funded by the Programa Presupuestario F003, project code CF 2019 265222 (CONAHCYT, México), granted to ASR. The funders had no role in study design, data collection and analysis, decision to publish, or preparation of the manuscript." Please include your amended statements within your cover letter; we will change the online submission form on your behalf. 6. Please take this opportunity to be sure you have met all of our guidelines for new species. When publishing papers that describe a new fungal taxon name, PLOS aims to comply with the requirements of the International Code of Nomenclature for algae, fungi, and plants (ICN). The following guidelines for publication in an online-only journal have been agreed such that any scientific fungal name published by us is considered effectively published under the rules of the Code. Please note that these guidelines differ from those for zoological nomenclature.Effective January 2012, ""the description or diagnosis required for valid publication of the name of a new taxon"" can be in either Latin or English. This does not affect the requirements for scientific names, which are still to be Latin.Also effective January 2012, the electronic PDF represents a published work according to the ICN for algae, fungi, and plants. Therefore the new names contained in the electronic publication of a PLOS ONE article are effectively published under that Code from the electronic edition alone, so there is no longer any need to provide printed copies.For proper registration of the new taxon, we require two specific statements to be included in your manuscript. A.)
In the Results section, the globally unique identifier (GUID), currently in the form of a Life Science Identifier (LSID), should be listed under the new species name, for example:Hymenogaster huthii. Stielow et al. 2010, sp. nov. [urn:lsid:indexfungorum.org:names:518624]You will need to contact either Mycobank or Index Fungorum to obtain the GUID (LSID).  B.)
In the Methods section, include a sub-section called ""Nomenclature"" using the following wording (this example is for taxon names submitted to MycoBank; please substitute appropriately if you have submitted to Index Fungorum and use the prefix http://www.indexfungorum.org/Names/NamesRecord.asp?RecordID= ):The electronic version of this article in Portable Document Format (PDF) in a work with an ISSN or ISBN will represent a published work according to the International Code of Nomenclature for algae, fungi, and plants, and hence the new names contained in the electronic publication of a PLOS ONE article are effectively published under that Code from the electronic edition alone, so there is no longer any need to provide printed copies.In addition, new names contained in this work have been submitted to MycoBank from where they will be made available to the Global Names Index. The unique MycoBank number can be resolved and the associated information viewed through any standard web browser by appending the MycoBank number contained in this publication to the prefix http://www.mycobank.org/MB/. The online version of this work is archived and available from the following digital repositories: [INSERT NAMES OF DIGITAL REPOSITORIES WHERE ACCEPTED MANUSCRIPT WILL BE SUBMITTED (PubMed Central, LOCKSS etc)].All PLOS ONE articles are deposited in PubMed Central and LOCKSS. If your institute, or those of your co-authors, has its own repository, we recommend that you also deposit the published online article there and include the name in your article.A complete explanation of our guidelines for publishing new species can be found on our website: http://www.plosone.org/static/guidelines#fungal Special Cases – Algae, plant fossils, etc.Please take this opportunity to be sure you have met all of our guidelines for new species. For submissions describing new species that do not have formal registries, please include a sub-section called “Nomenclature” in the Methods section using the following wording:The electronic version of this article in Portable Document Format (PDF) in a work with an ISSN or ISBN will represent a published work according to the International Code of Nomenclature for algae, fungi, and plants, and hence the new names contained in the electronic publication of a PLOS ONE article are effectively published under that Code from the electronic edition alone, so there is no longer any need to provide printed copies.The online version of this work is archived and available from the following digital repositories: PubMed Central, LOCKSS [author to insert names of any additional repositories where the work will be deposited].

Reviewers' comments:

Reviewer's Responses to Questions

**Comments to the Author**

1. Is the manuscript technically sound, and do the data support the conclusions?

Reviewer #1: Yes

2. Has the statistical analysis been performed appropriately and rigorously? 

Reviewer #1: N/A

3. Have the authors made all data underlying the findings in their manuscript fully available?

Reviewer #1: Yes

4. Is the manuscript presented in an intelligible fashion and written in standard English?

Reviewer #1: Yes

5. Review Comments to the Author

Reviewer #1: Review report of the research article titled “ Exophiala chapopotensis sp. nov., an extremotolerant black yeast from an oil-polluted soil in Mexico; phylophenetic approach to species hypothesis in the Herpotrichiellaceae family”

The article is very important and useful for the scientific community. Being a Medical Scientist, we are reporting novel species time to time. There is a guideline and protocol we follow when we claim a new species. I have gone through the sequencing-based evidence what the authors documented, and I am convinced that the genetic evidence is enough to prove that Exophiala chapopotensis is a novel species and different from other Exophila species.

Any organism when we start looking it, we first grow them to understand their morphology, when we confirm the genus under microscope, we then compare the morphology with other species with same genus. We then go for different phenotypic experiments example pH, temp stress etc. to determine the variations from other related species and finally we do the electron microscopy to confirm that. In this work, the authors quoted that ref., 17 they claimed that described the phenotypic details which already been published. When I have gone through that article, I didn’t find enough phenotypic evidence to prove that this could be a novel species.

Hence, I suggest the authors, try to isolate another one or two of same organism in possible. Describe the detail of phenotypic analysis of novel species with closely related organisms and finally electron microscopy to prove that it is a novel species. When they will provide the requested documents, the article can be reconsidered for another revision. I am giving here two ref. article, and it will help the authors to understand hoe to report a novel fungal species.

A) Aime et al. IMA Fungus (2021) 12:11

B) Singh S et.al, Frontiers in Cellular and Infection Microbiology, July 2021 | Volume 11 | Article 686120

6. PLOS authors have the option to publish the peer review history of their article (what does this mean?). If published, this will include your full peer review and any attached files.

Reviewer #1: No

---

## [Author Response · Author response to Decision Letter 0]

16 Nov 2023

Response to the editor and reviewers’ comments

We thank the editor and reviewers for their valuable comments and suggestions. See below a point-by-point response to all your comments:

General comments about the ‘Revised Manuscript’

The authors’ responses are in blue. The response to different elements of the reviewer's comments are separated by the symbol =>

Line numbers refer to the “Manuscript” (unmarked version). 

Journal Requirements:

Author’s Response: Thanks for this suggestion. We have ensured that the manuscript meets all PLOS ONE's style requirements this time.

2. In your Methods section, please provide additional information regarding the permits you obtained for the work. Please ensure you have included the full name of the authority that approved the field site access and, if no permits were required, a brief statement explaining why

Author’s Response: Thanks for this suggestion. No permit was needed for work or sampling. We have added a short statement explaining why on lines 108-109.

Author’s Response: We apologize for this confusion; we have now corrected that issue.

 "This research was partially funded by the Programa Presupuestario F003, project code CF 2019 265222 (CONAHCYT, México), granted to ASR. The funders had no role in study design, data collection and analysis, decision to publish, or preparation of the manuscript."

Author’s Response: Thanks for this suggestion. We have made the changes accordingly.

"MR. Ide-Pérez was supported by Consejo Nacional de Humanidades Ciencia y Tecnología (CONAHCYT), México (Grant number: 779850). We thank to IBT-UNAM and the program “Investigadoras e Investigadores por México” from the Consejo Nacional de Humanidades Ciencia y Tecnología (CONAHCYT), México, for supporting the Project 237. This research was partially funded by the Programa Presupuestario F003, project code CF 2019 265222, granted to ASR. Also, we thank to Dr. Alfonso Leija Salas for his help with the microscopy analysis."

"This research was partially funded by the Programa Presupuestario F003, project code CF 2019 265222 (CONAHCYT, México), granted to ASR. The funders had no role in study design, data collection and analysis, decision to publish, or preparation of the manuscript."

Author’s Response: We apologize for this confusion; we have now corrected that issue.

6. Please take this opportunity to be sure you have met all of our guidelines for new species. When publishing papers that describe a new fungal taxon name, PLOS aims to comply with the requirements of the International Code of Nomenclature for algae, fungi, and plants (ICN). The following guidelines for publication in an online-only journal have been agreed such that any scientific fungal name published by us is considered effectively published under the rules of the Code. Please note that these guidelines differ from those for zoological nomenclature.

Effective January 2012, ""the description or diagnosis required for valid publication of the name of a new taxon"" can be in either Latin or English. This does not affect the requirements for scientific names, which are still to be Latin.

Also effective January 2012, the electronic PDF represents a published work according to the ICN for algae, fungi, and plants. Therefore, the new names contained in the electronic publication of a PLOS ONE article are effectively published under that Code from the electronic edition alone, so there is no longer any need to provide printed copies.

For proper registration of the new taxon, we require two specific statements to be included in your manuscript.

 A.) In the Results section, the globally unique identifier (GUID), currently in the form of a Life Science Identifier (LSID), should be listed under the new species name, for example:

Hymenogaster huthii. Stielow et al. 2010, sp. nov. [urn:lsid:indexfungorum.org:names:518624]

You will need to contact either Mycobank or Index Fungorum to obtain the GUID (LSID). 

 B.) In the Methods section, include a sub-section called ""Nomenclature"" using the following wording (this example is for taxon names submitted to MycoBank; please substitute appropriately if you have submitted to Index Fungorum and use the prefix http://www.indexfungorum.org/Names/NamesRecord.asp?RecordID= ):

The electronic version of this article in Portable Document Format (PDF) in a work with an ISSN or ISBN will represent a published work according to the International Code of Nomenclature for algae, fungi, and plants, and hence the new names contained in the electronic publication of a PLOS ONE article are effectively published under that Code from the electronic edition alone, so there is no longer any need to provide printed copies.

In addition, new names contained in this work have been submitted to MycoBank from where they will be made available to the Global Names Index. The unique MycoBank number can be resolved and the associated information viewed through any standard web browser by appending the MycoBank number contained in this publication to the prefix http://www.mycobank.org/MB/. The online version of this work is archived and available from the following digital repositories: [INSERT NAMES OF DIGITAL REPOSITORIES WHERE ACCEPTED MANUSCRIPT WILL BE SUBMITTED (PubMed Central, LOCKSS etc)].

All PLOS ONE articles are deposited in PubMed Central and LOCKSS. If your institute, or those of your co-authors, has its own repository, we recommend that you also deposit the published online article there and include the name in your article.

A complete explanation of our guidelines for publishing new species can be found on our website: http://www.plosone.org/static/guidelines#fungal

Special Cases – Algae, plant fossils, etc.

Please take this opportunity to be sure you have met all of our guidelines for new species. For submissions describing new species that do not have formal registries, please include a sub-section called “Nomenclature” in the Methods section using the following wording:

The electronic version of this article in Portable Document Format (PDF) in a work with an ISSN or ISBN will represent a published work according to the International Code of Nomenclature for algae, fungi, and plants, and hence the new names contained in the electronic publication of a PLOS ONE article are effectively published under that Code from the electronic edition alone, so there is no longer any need to provide printed copies.

The online version of this work is archived and available from the following digital repositories: PubMed Central, LOCKSS [author to insert names of any additional repositories where the work will be deposited].

Author’s Response: Thanks for this important reminder. We have included the two specific statements in the manuscript:

A.) The globally unique identifier (GUID), Life Science Identifier (LSID) in the Results section as:

Exophiala chapopotensis. Ide-Pérez et al. 2023, sp. nov. (Fig 4 A-F). Fungal Names no. FN 571584.

urn:lsid:nmdc.cn:fungalnames:571584

B.) The sub-section "Nomenclature" in the Methods, with the appropriate text (lines 198-211).

Response to the Reviewer Comments

Reviewer #1: 

The article is very important and useful for the scientific community. Being a Medical Scientist, we are reporting novel species time to time. There is a guideline and protocol we follow when we claim a new species. I have gone through the sequencing-based evidence what the authors documented, and I am convinced that the genetic evidence is enough to prove that Exophiala chapopotensis is a novel species and different from other Exophila species.

Author’s Response: Thank you for your comments and observations. We believe that genetic-genomic information is of paramount importance in present-day taxonomy. Some already talk about taxogenomics, although we are more akin to an integrative approach: genetics, genomics, phenotype and canonical models of speciation together with conceptual frameworks such as the phylophenetic one claimed in this manuscript.

Any organism when we start looking it, we first grow them to understand their morphology, when we confirm the genus under microscope, we then compare the morphology with other species with same genus. We then go for different phenotypic experiments example pH, temp stress etc. to determine the variations from other related species and finally we do the electron microscopy to confirm that. In this work, the authors quoted that ref., 17 they claimed that described the phenotypic details which already been published. When I have gone through that article, I didn’t find enough phenotypic evidence to prove that this could be a novel species.

Hence, I suggest the authors, try to isolate another one or two of same organism in possible. Describe the detail of phenotypic analysis of novel species with closely related organisms and finally electron microscopy to prove that it is a novel species. When they will provide the requested documents, the article can be reconsidered for another revision. I am giving here two ref. article, and it will help the authors to understand hoe to report a novel fungal species.

A) Aime et al. IMA Fungus (2021) 12:11

B) Singh S et.al, Frontiers in Cellular and Infection Microbiology, July 2021 | Volume 11 | Article 686120 

Author’s Response: Thank you for your pertinent observations.

=>Our thesis is that any individual hypothesis alone (e.g. phenotypic traits, phylogenetic traits, etc.) has little predictive value for accepting or rejecting new microbial species. That’s why we use a robust conceptual framework, through the integrative analysis of multiple working hypotheses that test the major hypothesis of speciation. Namely, under the phylophenetic concept claimed in this manuscript: 1-Genomic coherence, 2-Monophily or phylogenetic transition, 3- Evolutionary-molecular speciation signatures and finally 4-Transitional phenotypic characters. We hold that a modern taxonomic exercise is not about to find “phenotypic evidence to prove that this could be a novel species” but to detect the signatures of phenotypic transitions that support the speciation model. Phenotypic plasticity is common in microorganisms, assuming that the phenotype contains ‘enough evidence to prove novel species’ is a such premise that if it were true, then the problem of the species would already be solved. In line with all the evidence for the first three hypotheses, we already offer in the text a series of phenotypic traits that represent true biological transitions signatures that diagnose the proposed species (section Micromorphology, phenotypic and other transitive characteristic of LBMH1013), namely: 

The strain LBMH1013 differs from the closely related E. nidicola by its larger aseptate conidia (lines 362-363)

The conidial septation is an important transition criterion relating to developmental biology; It has a long history among systematic specialists since it is an essential process in the ontogeny of conidial fungi, in development and conidium completion. The contrasts in this character imply totally differential development processes of the conidium structure (Boerema, G.H, & Bollen, G.J. (1975). Persoonia, 8(2), 111–144; Cole, G. T. (1986). Microbiological reviews, 50(2), 95-132; Alves et al. (2008). Fungal diversity, 28, 1-13) 

The absence of growth at 40 °C is a distinguishing character for LBMH1013, also at 37° the growth is severely disrupted (17), which is a distinctive feature of E. dermatitidis and E. heteromorpha (lines 363-366)

In many black yeasts, the ability to grow above 37°C is invariably linked to their pathogenic and invasive potential. It constitutes an adaptive criterion with important evolutionary implications since it implies -in many cases- potential morphological switching and the formation of emerging ecotypes. The temperature barrier remains one the key defense mechanism of warm-blooded organisms against fungal infections (along with adaptive and innate immunity). The phenotypic growth dichotomy between this boundary separates biological contexts according to their potential to conquer new niches (ecological premise of the species) (Casadevall A, Kontoyiannis DP, Robert V. On the Emergence of Candida auris: Climate Change, Azoles, Swamps, and Birds. mBio. 2019 Jul 23;10(4):e01397-19. doi: 10.1128/mBio.01397-19. PMID: 31337723; PMCID: PMC6650554).. 

Additionally, LBMH1013 only tolerates ~5.84% NaCl, unlike E. heteromorpha in which viability has been observed at 10% (lines 366-367)

Tolerance to high concentrations of salts is also a differential adaptive criterion related to success under stress conditions and in black yeasts it has been associated with the capacity for morphological switching to yeast morphotypes, with a greater capacity for cell division, etc. It also responds to the ecological premise of the species and has differentiated genetic bases.

…the strain LBMH1013 seems to be heterothallic, since the genome contains markers for only one of the MAT sexual idiomorphs (MAT1-1-4 and MAT1-1-1 (alpha-box)) (lines 397-399)

One of the most robust phenotypic criteria to approximate the biological concept in fungi. The fact that most representatives of Exophiala are heterothallic separates them conclusively from another historically and closely related genus, Capronia, in many cases mistakenly considered to be its teleomorph (Teixeira et al. Exploring the genomic diversity of black yeasts and relatives (Chaetothyriales, Ascomycota). Stud Mycol. 2017;86: 1–28).

This set of phenotypic characters contrast the Transitional phenotypic characters’ hypothesis as a result of speciation. The alternative hypothesis (there are no phenotypic signatures that support speciation) is evidently falsified with the description of these observations, which involve elements of the biology of fungal development, two approaches to the ecological premise of the species and evidence on the determinants of sexuality. 

=>However, we agree with the reviewer in the importance of describe a more detailed phenotypic analysis. We have incorporated a new table with the metabolic responses of LBMH1013 to different carbon sources, growth at different temperatures, pH and salinity (Table 4). A detailed description is shown in lines 378-387 and one supplementary figure was also included (S2 Fig). The proper methodology description of this part was added in lines 114-120. It was necessary to cite additional works in this review: [10,56,65–70,57–64]. With this exercise we confirmed other variations from related species, such as growth on lactose and urease production with E. heteromorpha and glucosamine utilization and glucose fermentation with E. dermatitidis. 

=>We have not been successful in obtaining new isolates of the same species. Our attempts from new samples in the same area have been unsuccessful so far. This may be because, in all environmental ecosystems ecological successions dominate the microbiodiversity landscape. Unlike microorganisms from clinical microenvironments (generally surrounded by nutritionally rich niches and convenient physical barriers to development), environmental isolates face fluctuating environmental and nutritional conditions that cause successions in composition and abundance. Especially in this case that deals with an isolate from an oligotrophic environment, in semi-arid terrain historically impacted with hydrocarbons.

=>Although we recognize the importance of multi-isolates and electron microscopy, the execution of these statements neither proves nor disproves that this is a new species. They are rather methodological approaches, that although they can improve the resolution of certain aspects of the proposal, they do not establish differential selection criteria for the species hypothesis. E.g. if electron microscopy was a necessary condition, many species would not have been defined today, especially because the morphological structures of the mycelium and the spore are fairly approachable with classical optical microscopy techniques. We made new images and modified Fig 4 in its panels D-F to better illustrate the microscopic characteristics of this taxon. We tried to get images using an Environmental Scanning Electronic Microscopy, however the acquired images were not entirely decisive but nevertheless we include these in the document to be consulted although we did not include them in the text of the paper:

.

=>We appreciate your recommendations of the two references on how to report a new fungal species. We have followed the Aime et al. IMA Fungus 12, 11 (2021) checklist for publishing new species. All required actions are met in this description and most of the desired actions. We regret not taking the second manuscript into account, since we consider that it is focused on a specimen of clinical origin where only the phylogenetic and phenotypic hypotheses are verified. Our proposal contains a more complete and testable conceptual framework.

---

## [Editor Report · Decision Letter 1]

2 Jan 2024

Exophiala chapopotensis sp. nov., an extremotolerant black yeast from an oil-polluted soil in Mexico; phylophenetic approach to species hypothesis in the Herpotrichiellaceae family

PONE-D-23-27467R1

Dear Dr. Sánchez Reyes,

We’re pleased to inform you that your manuscript has been judged scientifically suitable for publication and will be formally accepted for publication once it meets all outstanding technical requirements.

Kind regards,

Rajeev Singh

Academic Editor

PLOS ONE
---

## [Editor Report · Acceptance letter]

25 Jan 2024

PONE-D-23-27467R1 

PLOS ONE

Dear Dr. Sánchez Reyes, 

I'm pleased to inform you that your manuscript has been deemed suitable for publication in PLOS ONE. Congratulations! Your manuscript is now being handed over to our production team.

Kind regards, 

on behalf of

Dr. Rajeev Singh 

Academic Editor

PLOS ONE